# Shared Steering Control for Lane Keeping and Obstacle Avoidance Based on Multi-Objective MPC

**DOI:** 10.3390/s21144671

**Published:** 2021-07-08

**Authors:** Yang Liang, Zhishuai Yin, Linzhen Nie

**Affiliations:** 1School of Automotive Engineering, Wuhan University of Technology, Wuhan 430070, China; 250625@whut.edu.cn (Y.L.); linzhen_nie@whut.edu.cn (L.N.); 2Hubei Key Laboratory of Advanced Technology for Automotive Components, Wuhan University of Technology, Wuhan 430070, China; 3Hubei Collaborative Innovation Center for Automotive Components Technology, Wuhan University of Technology, Wuhan 430070, China; 4Hubei Research Center for New Energy & Intelligent Connected Vehicle, Wuhan University of Technology, Wuhan 430070, China

**Keywords:** shared control, model predictive control, situation assessment

## Abstract

This paper presents a shared steering control framework for lane keeping and obstacle avoidance based on multi-objective model predictive control. One of the control objectives is to track the reference trajectory, which is updated continuously by the trajectory planning module; whereas the other is to track the driver’s current steering command, so as to consider the driver’s intention. By adding the two control objectives to the cost function of an MPC shared controller, a smooth combination of the commands of the driver and the automation can be achieved through the optimization. The authority of the driver and the automation is allocated by adjusting the weights of the objective terms in the cost function, which is determined by the proposed situation assessment method considering the longitudinal and lateral risks simultaneously. The results of the CarSim-Matlab/Simulink joint simulations show that the proposed shared controller can assist the driver to complete the tasks of lane keeping and obstacle avoidance smoothly while maintaining a good level of vehicle stability.

## 1. Introduction

In the past decade, the development of autonomous vehicles has received a great amount of attention and brought considerable benefits. Although full automation under all driving circumstances remains an extremely challenging task due to numerous technical limitations and legal issues [1,2], vehicles equipped with advanced driver assistance systems (ADAS), which fit into the level of partial automation (PA) or below as defined by SAE [3], have already been proven to reduce traffic accidents [4] while lowering the burden on drivers [5]. Industrial entities and researchers are, therefore, motivated to exploit benefits of higher levels of automation which involve more frequent and tight human–automation cooperation.

Early studies proposed to switch the control authority back and forth between the driver and the automation system based on the driver’s attentiveness [6] and/or the capacity of the automation system [7]. However, some studies have shown that it is in fact challenging for the driver to resume control over the vehicle due to issues such as over-trust in automation [8], or lack of situation awareness [9].

Therefore, it has been proposed to keep the driver in the loop while allowing automation systems to cooperate with drivers so as to improve driving safety [1]. Shared control, which allows the driver and the automation system to drive the vehicle simultaneously by combining control actions from both, is considered as a feasible and promising solution to the transition from low levels of autonomy to conditional or even high automation [10].

A shared control scheme mainly involves driver–automation interactions at the tactical level and the operational level. The tactical level focuses on the decision-making process of when and where the driver or the automation system should intervene. Some studies propose to shift the control authority based on the driver’s intention, which is either simulated with driving behavior models [11,12] or interpreted via monitoring the driver’s state and tracking the driver’s control actions. Bencloucif, A.M. et al. [13] propose a cooperative trajectory planner, which updates the reference trajectory according to the driver’s state and control actions. Another category of shared control strategies at the tactical level is designed on the basis of driving risk assessment. Li, M. et al. [14] define the positions of 6S, 4S and 2S of TTC (time to collision) as representative nodes of driving risk, and build the relationship between driving risk and control authority into a lookup table. Liu, R. et al. [15] propose to characterize driving risk with time margin (TM) and TTC combined, and allocate the control authority accordingly with explicit logistics. Some recent works emphasize the importance of taking the factor of the driver, the vehicle, and the driving environment all into account to achieve better driving comfort and safety. Chouki, S. et al. [16] evaluate the state of the driver through the driver monitoring system (DMS) and design the control strategy in combination with the risk of lane departure. Li, M. et al. [17] design the consistent fuzzy controller, the advanced inconsistent fuzzy controller, and the lagged inconsistent fuzzy controller, respectively, according to the intention consistency between the driver and the automation.

In terms of human–vehicle interactions at the operational level, two frameworks based on different control mechanisms have been proposed: (1) Haptic shared control [11,12,13,18,19,20,21], in which the driver’s commands are directly applied to the vehicle through a mechanical coupled steering system while the automation system intervenes by applying additional torques. This control framework has shown its advantages in continuous haptic interaction between the driver and the automation [22]. However, some researches have shown that the neuromuscular characteristics of the driver should be considered when applying the additional torque, otherwise both driving safety and comfort could be negatively impacted [23]. Moreover, unconscious resistance from the driver may be caused when unexpected torques are applied in emergency scenarios, such as obstacle avoidance [24]. (2) Input-mixing systems, in which the controller has the final authority and can modify the driver’s commands before they are applied to the vehicle through the steer-by-wire system. Under such a control framework, the automation can modify the driver’s input arbitrarily and has the final control authority [25]. The final steering command could be an adaptively weighted fusion of the commands of the driver and the automation system [14,15,16,17,26,27,28,29], or a transformation from the driver’s commands with a designed algorithm [30]. Erlien, S.M. et al. [31] designed a shared controller in the framework of MPC (Model Predictive Control) and incorporated the driver’s steering commands into the cost function of the MPC formulation, so that matching the driver’s command becomes a control objective. The driver’s steering command is corrected by the automation system only when the vehicle is outside the safety envelopes defined by the vehicle handling limits and spatial limitations formed by lane boundaries and obstacles. Though, it is worth noting that with fixed safety envelopes, the intervention from the automation system may be untimely under highly dynamic conditions where the driving risk evolves rapidly.

Inspired by the work of Erlien, S.M. et al. [31], this paper formulates the shared control between the driver and the automation as a multi-objective optimization problem which is a trade-off between matching driver command and tracking reference trajectory. Model Predictive Control, which is suitable for multi-objective optimization problems with constrains and has the advantages of prediction, is now adopted by many studies of shared control [21,25,26,27,28,29,30,31]. Thus, we propose a two-level framework of an MPC-based shared controller for lane keeping and obstacle avoidance, as shown in Figure 1. Firstly, to obtain a collision-free trajectory in real-time, the information of the obstacle is added to the cost function of an NMPC (Nonlinear Model Predictive Control)-based trajectory planner. Secondly, a situation assessment method, considering the lateral and longitudinal risks, is designed to determine the weights of the two optimization objectives, so that timely responses at the operational level can be achieved under challenging driving conditions. Finally, with matching the driver’s commands and tracking the planned obstacle avoidance trajectories both taken as the optimization objectives, the proposed controller manages to keep the driver in the loop and avoids having to interpret the driver’s intentions. By dynamically adjusting the weights of the two optimization objectives in the MPC cost function, rather than blending commands of the driver and the automation system, the authority allocation can be implemented smoother in an optimal controller. The main contribution of this paper is as follows:A new shared control framework based on MPC is proposed, which includes an NMPC-based trajectory planning module, a situation assessment module considering both lateral and longitudinal risks, and a shared control module based on multi-objective MPC.A shared control scheme based on multi-objective MPC is proposed, in which matching the driver’s commands and tracking the reference trajectories are both taken as the optimization objectives, and the authority of the driver and the automation are allocated by adjusting the weights of the two objectives, which are determined by the situation assessment module.

## 2. Vehicle Model

In this section, a vehicle dynamics model, which is used as the prediction model of the MPC-based shared controller, is introduced. The parameters used in the vehicle model are presented in Table 1.

Computation time is a major challenge in the design of an MPC, in which the complexity of the prediction model affects a lot. However, the complexity and the accuracy of the prediction model are two conflictive aspects. Many studies have proposed to simplify the prediction model to achieve a trade-off between accuracy and complexity. The bicycle model, which has been proved to be effective in achieving a trade-off between modelling accuracy and computation efficiency [32], was adopted as the prediction model of the MPC-based controller in this study. Since only the lateral control of the vehicle was considered, the longitudinal speed of the vehicle was set to be unchanged throughout the control process, and only the front wheel angle was considered as the input of the vehicle dynamics model. Meanwhile, it was assumed that the vehicle was equipped with a steer-by-wire system so that the mechanical connection between the steering wheel and the front wheel can be decoupled.

As shown in Figure 2, a dynamics model of the bicycle vehicle was established with three degrees of freedom, including longitudinal, lateral, and yaw motion. According to Newton’s second law, the following force balance equations can be obtained along the *X*-axis, *Y*-axis, and *Z*-axis:(1)$$\begin{array}{l} m{\dot{v}}_{x}=mv_{y}r+2\left(F_{l,f}\cos\delta_{f}-F_{c,f}\sin\delta_{f}\right)+2F_{l,r}\\ m{\dot{v}}_{y}=-mv_{x}r+2\left(F_{l,f}\sin\delta_{f}+F_{c,f}\cos\delta_{f}\right)+2F_{c,r}\\ I_{z}\dot{r}=2a\left(F_{l,f}\sin\delta_{f}+F_{c,f}\cos\delta_{f}\right)-2bF_{l,r} \end{array}$$
where $v_{y}\;\mathrm{and}\;v_{x}$, respectively, represent the lateral and longitudinal speed of the vehicle in the body-fixed coordinate system.

The tire force in this study is simplified as follows [33]:(2)$$F_{c,i}=C_{c,i}\alpha_{i},\;i=f,r$$
(3)$$F_{l,i}=C_{l,i}\kappa_{i},\;i=f,r$$

Assuming that the vehicle was equipped with an Antilock Brake System (ABS) system, the slip ratio $\kappa_{i}$ was identified. When the slip angle is relatively small, the tire cornering angle can be obtained with the following formula [34]:(4)$$\alpha_{f}=\frac{\dot{y}+a\dot{\vartheta }}{\dot{x}}-\delta_{f}$$
(5)$$\alpha_{r}=\frac{\dot{y}-b\dot{\vartheta }}{\dot{x}}$$

The state variables in the vehicle coordinate system are converted to those in the inertial coordinate system as:(6)$$\dot{X}=v_{x}\cos\vartheta -v_{y}\sin\vartheta \;\dot{Y}=v_{x}\sin\vartheta +v_{y}\cos\vartheta$$

## 3. Dynamic Trajectory Planning Based on NMPC

A dynamic trajectory planner is designed to plan the obstacle avoidance trajectory in real-time, as the reference trajectory for which the shared controller tracks. To reduce the computational complexity of the NMPC, the motion of the vehicle is described as a particle [35]:(7)$${\dot{v}}_{y}=\gamma_{y}\;{\dot{v}}_{x}=0\;\dot{\vartheta }=\frac{\gamma_{y}}{v_{x}}$$

By discretizing Equations (6) and (7) with the Euler method, the prediction equation of the trajectory planner can be obtained:(8)$$\xi_{t}\left(s+i\left|s\right.\right)=\Gamma \left[\xi_{t}\left(s+i-1\left|s\right.\right),u_{t}\left(s+i-1\left|s\right.\right)\right]\;\eta_{t}=J_{t}\xi_{t}\;i=1,2,\ldots P_{t}\;,\;\;J_{t}=\left[\begin{array}{ccccc} 0 & 0 & 0 & 0 & 1 \end{array}\right]$$
where, Γ represents the discrete form of Equations (6) and (7), $\xi_{t}=\left[\begin{array}{ccccc} v_{x} & v_{y} & \vartheta & X & Y \end{array}\right]$ represents the vehicle states in the trajectory planning controller, $P_{t}$ is the prediction horizon, and $\xi_{t}\left(s+i\left|s\right.\right)$ represents the predicted vehicle state at step s+i. The lateral acceleration of the vehicle $\gamma_{y}$ is chosen as the control input, and $u_{t}$ and $u_{t}\left(s+i\left|s\right.\right)$ represent the control input at time step s+i. To reduce the amount of calculation, it is assumed that the control input outside the control horizon remains unchanged:(9)$$u_{t}\left(s+i\left|s\right.\right)=u_{t}\left(s+C_{t}\left|s\right.\right),i=C_{t}+1,\ldots P_{t}$$
where, $C_{t}$ is the control horizon, and $C_{t}\leq P_{t}$. The goal of dynamic trajectory planning is to avoid obstacles while minimizing the error between the planned trajectory and the global reference trajectory, so the mathematical expression of the objective function is expressed as follows:(10)$$\begin{array}{c} \underset{u_{t}}{min}\overset{P_{t}}{\underset{i=1}{\sum }}||\eta_{t}\left(s+i\left|s\right.\right)-\eta_{gr}{\left(s+i\left|s\right.\right)||}_{Q_{t}}^{2}+\overset{C_{t}}{\underset{i=1}{\sum }}||u_{t}{\left(s+i\left|s\right.\right)||}_{R_{t}}^{2}\\ +\overset{M_{obs}}{\underset{j=1}{\sum }}J_{obs,j}\\ s.t.\;u_{min}<u\left(s+i|s\right)<u_{max},i=1,2\ldots C_{t} \end{array}$$
where, $\eta_{gr}\left(s+i\left|s\right.\right),i=1,2,\ldots ,\;P_{t}$ represents the global reference trajectory in the prediction horizon, $Q_{t}\;\mathrm{and}\;R_{t}$ represent the weight matrix of controlled outputs and inputs, $M_{obs}$ represents the number of obstacle points, and $J_{obs,j}$ represents the obstacle avoidance function of the jth obstacle point, which is inversely proportional to the Euclidean distance between the COG (Center Of Gravity) and the obstacle, and is proportional to the vehicle speed. $J_{obs,j}$ is a function designed to produce a penalty for obstacle avoidance. The mathematical expression is as follows:(11)$$J_{obs,j}=\frac{vK_{obs}}{{\left(x_{c}-x_{obs,j}\right)}^{2}+{\left(y_{c}-y_{obs,j}\right)}^{2}+\zeta }$$
where, v denotes the vehicle speed, $K_{obs}$ denotes the weight of the obstacle avoidance function, $\left(x_{c},y_{c}\right)$ is the coordinate of the COG, $\left(x_{obs,j},y_{obs,j}\right)$ represents the coordinates of obstacle points, and ζ is a very small number to prevent the denominator from being 0. The diagram of the obstacle avoidance function is shown in Figure 3.

To lower the computational overhead, the reference trajectory is produced with the fifth-order polynomial curve fitting, whose mathematical expression is as follows:(12)$$Y_{ref}=a_{0}+a_{1}X^{1}+a_{2}X^{2}+a_{3}X^{3}+a_{4}X^{4}+a_{5}X^{5}$$
where, $a_{0},a_{1},a_{2},a_{3},a_{4}\;\mathrm{and}\;a_{5}$ are the parameters to be fitted, $Y_{ref}$ can be the reference lateral position $y_{ref}$ or the reference heading angle $\vartheta_{ref}$, and X indicates the longitudinal position of the vehicle.

## 4. MPC-Based Shared Controller

The MPC-based shared controller is designed to track the reference trajectory while matching the driver’s steering commands whenever safe. The trade-off between the two objectives is achieved through adjusting the weights, which are determined on the basis of a real-time assessment of the driving situation.

### 4.1. Situation Assessment

In this paper, we chose Time to Lane Crossing (TLC) [36] and Time to Collision (TTC) [37] as two factors of the driving risk assessment. TLC is defined as the time remaining for the front wheels to reach the border of the current lane if the vehicle maintains the current lateral acceleration. The expression of TLC is as follows:(13)$$TLC=\frac{-v_{y}+\sqrt{v_{y}^{2}+2\gamma_{y}y_{ll}}}{\gamma_{y}}$$
where $y_{ll}$ can be the distance from the left (or right) front wheel to the left (or right) lane boundary. The measurement of the lateral vehicle speed $v_{y}$ is quite challenging for it is not available from a conventional sensor. However, previous studies have proposed to estimate the lateral vehicle speed based on measurable parameters [36,38]. Therefore, it is assumed that the lateral vehicle speed $v_{y}$ can be obtained by estimation.

As shown in Figure 4, TTC is defined as the time remaining for the vehicle to collide with the obstacle if the vehicle maintains the current velocity. The expression of TTC is as follows:(14)$$TTC=\left\{\begin{array}{l} \frac{\left||{\vec{p}}_{c,o}|\right|}{vcos(\vec{v},{\vec{p}}_{c,o})},cos(\vec{v},{\vec{p}}_{c,o})>0\\ \infty \;\;\;,cos(\vec{v},{\vec{p}}_{c,o})<0 \end{array}\right.\;{\vec{p}}_{c,o}=\left(x_{o}-x_{c},y_{o}-y_{c}\right)$$
where, $\left(x_{o}\;\mathrm{and}\;y_{o}\right)$ denote the coordinates of the obstacle, $\left(x_{c}\;\mathrm{and}\;y_{c}\right)$ denote the coordinates of the vehicle, and $p_{c,o}$ is the vector of the mass center of the vehicle pointing to the mass center of the obstacle.

In this paper, TLC and TTC were both used for the situation assessment. When there is no collision risk, the authority of the controller is adjusted according to TLC for the purpose of lane keeping. If TLC reaches the low-risk threshold, the control authority transits from matching the driver’s commands to tracking the global reference trajectory; When obstacle avoidance becomes the prominent and pressing task, TTC is used as the risk indicator, and the control authority transits from matching the driver’s commands to tracking the dynamically planned obstacle avoidance trajectory. As suggested in previous studies [39,40], 2 s and 4 s are used as the high-risk threshold and the low-risk threshold of TTC, respectively, while the high-risk threshold of TLC is defined as [41]:(15)$$TLC_{min}=\frac{2v_{x}\vartheta }{\mu g}+t_{d}$$
where, $t_{d}$ is the total response time of the actuator and the driver, and we chose $t_{d}$ = 1 s in this paper. In addition, the low-risk threshold of TLC is defined as $TLC_{max}=2TLC_{min}$. Considering the conflict between the driver and the automation in the scenario of lane keeping, a threshold of the driver’s steering command $\sigma_{lim}$ is introduced when evaluating the lane departure risk. When the risk of lane departure increases and the driver input continues to exceed this threshold, we consider that the lane keeping system needs to be deactivated and the driver has the control of the vehicle. In addition, we set $\sigma_{lim}\;$ = 2° in this study. The weight factor is obtained by the following formula:(16)$$\sigma =\left\{\begin{array}{cc} 0 & \begin{array}{c} ,(TTC>TTC_{max}\;\;\;and\;\;TLC>TLC_{max})\\ \;or\;\left(TLC\langle TTC_{max}\;\;and\;\;\sigma_{d}\rangle \sigma_{lim}\right) \end{array}\\ \frac{TLC-TLC_{min}}{TLC_{max}-TLC_{min}}\; & \begin{array}{c} ,TTC>TTC_{max}\;and\;\;TLC_{min}<TLC<TLC_{max}\\ and\;\sigma_{d}\leq \sigma_{lim} \end{array}\;\\ \frac{TTC-TTC_{min}}{TTC_{max}-TTC_{min}} & ,TTC_{min}<TTC<TTC_{max}\\ 1 & ,\;TTC<TTC_{min}\;\;or\;(TLC<TLC_{min\;}and\;\sigma_{d}\leq \sigma_{lim})\; \end{array}\right.$$
where, $TLC_{min}$ and $TTC_{min}$ represent the high-risk threshold of TLC and TTC, and $TLC_{max}$ and $TTC_{max}$ represent the low-risk threshold of TLC and TTC, respectively.

### 4.2. Design of MPC Shared Controller

After discretizing the vehicle dynamics model established in Section 2 with a fixed sampling time $T_{s}$, the following expression is obtained:(17)$$\begin{array}{c} \xi_{s}\left(s+1\right)=f\left(\xi_{s}\left(s\right),\Delta \delta_{f}\left(s\right)\right)\\ \xi_{s}\left(s\right)={\left[\begin{array}{ccccccc} v_{y}\left(s\right) & v_{x}\left(s\right) & \vartheta \left(s\right) & r\left(s\right) & Y\left(s\right) & X\left(s\right) & \delta_{f}\left(s-1\right) \end{array}\right]}^{T}\\ \Delta \delta_{f}\left(s\right)=\delta_{f}\left(s\right)-\delta_{f}\left(s-1\right) \end{array}$$
where, s represents the current step, and $\Delta \delta_{f}$ is the control input. To reduce the computational complexity so as to improve the real-time performance of the controller, Equation (17) is linearized around an operating point:(18)$$\begin{array}{c} \xi_{s}\left(s+1\right)=G_{s}\xi_{s}\left(s\right)+H_{s}\Delta \delta_{f}\left(s\right)\\ \eta_{s}\left(s+1\right)=J_{s}\xi_{s}\left(s\right) \end{array}$$
where,
(19)$$G_{s}=\left[\begin{array}{cc} A_{s} & B_{s}\\ 0_{1\times 6} & I \end{array}\right],H_{s}=\left[\begin{array}{c} B_{s}\\ I \end{array}\right],J_{s}=\left[\begin{array}{ccccccc} 0 & 0 & 0 & 0 & 1 & 0 & 0\\ 0 & 0 & 1 & 0 & 0 & 0 & 0 \end{array}\right]$$

The control objectives of MPC shared controller are to track the reference trajectories and match the driver’s steering commands. A dynamic and smooth shift of human–machine control authorities is accomplished by adjusting the weight matrix of the two control objectives.
(20)$$min\overset{P_{s}}{\underset{i=1}{\sum }}\left\{||\eta_{s}\left(\left.s+i\right|s\right)-\eta_{ref}{\left(\left.s+i\right|s\right)||}_{\sigma Q}^{2}+||\delta_{f}\left(s\left|s\right.\right)-\delta_{d}||{}_{\left(1-\sigma \right)S}^{2}\right\}+\overset{C_{s}-1}{\underset{i=0}{\sum }}||\Delta \delta_{f}{\left(\left.s+i\right|s\right)||}_{R}^{2}$$
(21)$$\begin{array}{c} s.t.\;\delta_{f,\min}\leq \delta_{f}\leq \delta_{f,\max}\\ \;\;\;\;\;\;\;\;\;\;\;\;\;\;\Delta \delta_{f,\min}\leq \Delta \delta_{f}\leq \delta_{f,\max} \end{array}$$
where, $\eta_{s}$ represents the output of the prediction model, $\eta_{ref}$ represents the reference output, $P_{s}$ and $C_{s}$ represent the prediction horizon and the control horizon in the shared controller, respectively, $\delta_{d}$ represents the driver’s current steering command, and Q, S and  R, respectively, represent the weight matrix of the trajectory tracking, driver’s command, and control inputs. σ is the shared control weight factor, which falls within the interval (0,1). When σ is large, the objective of the tracking reference trajectories prevails; on the contrary, the driver dominates the control. Equation (21) represents the constraints on the control input and control increment.

Assuming that the front wheel angle output by the driver remains unchanged in the control horizon and define:$$\begin{array}{c} \Delta U\left(s\right)={\left\{\begin{array}{c} \Delta \delta_{f}\left(s\right)\\ \Delta \delta_{f}\left(s+1\right)\\ \vdots \\ \Delta \delta_{f}\left(s+C_{s}\right) \end{array}\right\}}_{C_{s}\times 1}\;\;\;\;\;\;U_{d}\left(k\right)={\left\{\begin{array}{c} \delta_{d}\left(s\right)\\ \delta_{d}\left(s\right)\\ \vdots \\ \delta_{d}\left(s\right) \end{array}\right\}}_{C_{s}\times 1}\\ M={\left\{\begin{array}{cccc} 1 & 0 & \cdots & 0\\ 1 & 1 & \cdots & 0\\ \vdots & \vdots & \ddots & \vdots \\ 1 & 1 & \cdots & 1 \end{array}\right\}}_{C_{s}\times C_{s}}\;\;\;\;\;\;\;\;S_{e}={\left(\begin{array}{cccc} S & 0 & \cdots & 0\\ 0 & S & \cdots & 0\\ \vdots & \vdots & \ddots & \vdots \\ 0 & 0 & \cdots & S \end{array}\right)}_{C_{s}\times C_{s}} \end{array}$$

The objective function in Equation (20) is transformed into the following quadratic programming (QP) form:(22)$$\begin{array}{c} \underset{\Delta U\left(s\right),\varepsilon }{min}\frac{1}{2}{\left[\begin{array}{c} \Delta U\left(s\right)\\ \varepsilon \end{array}\right]}^{T}H_{e}\left[\begin{array}{c} \Delta U\left(s\right)\\ \varepsilon \end{array}\right]+G_{e}\left[\begin{array}{c} \Delta U\left(s\right)\\ \varepsilon \end{array}\right]+P_{e}\\ s.t.\left[\begin{array}{cc} I & 0\\ -I & 0\\ M & 0\\ -M & 0 \end{array}\right]\left[\begin{array}{c} \Delta U\left(s\right)\\ \varepsilon \end{array}\right]\leq \left[\begin{array}{c} \Delta \delta_{f,max}\times ones\left(C_{s},1\right)\\ -\Delta \delta_{f,min}\times ones\left(C_{s},1\right)\\ \delta_{f,max}\times ones\left(C_{s},1\right)\\ -\delta_{f,min}\times ones\left(C_{s},1\right) \end{array}\right] \end{array}$$
where,
(23)$$\begin{array}{c} H_{e}=\left[\begin{array}{cc} 2\left(\Theta_{s}{}^{T}Q\Theta_{s}+R+M^{T}SM\right) & 0_{C_{s}\times 1}\\ 0_{1\times C_{s}} & \rho \end{array}\right]\;\;\;\;\;\;\;\;\;\;\;\;\;\;\;\;\;\;G_{e}=\left[\begin{array}{cc} 2(E\left(s{)}^{T}Q\Theta_{k}+U\left(s-1\right)\mathrm{SM}-U_{d}{}^{T}S_{e}M\right) & 0 \end{array}\right]\\ P_{e}=E{\left(s\right)}^{T}QE\left(s\right)-2U_{d}{}^{T}SU\left(s-1\right)+U_{d}{}^{T}SU_{d},\;E\left(s\right)=\Gamma_{s}\xi \left(s\right)-Y_{ref}\\ \Gamma_{s}=\left[\begin{array}{c} J_{s}G_{s}\\ J_{s}G_{s}G_{s+1}\\ \vdots \\ J_{s}\overset{P_{s}-1}{\underset{i=0}{\prod }}G_{s+i} \end{array}\right],\;\Theta_{s}=\left[\begin{array}{cccc} J_{s}H_{s} & 0 & \cdots & 0\\ J_{s}G_{s}H_{s} & J_{s}H_{s} & \cdots & 0\\ \vdots & \vdots & \ddots & \vdots \\ J_{s}\overset{P_{s}-1}{\underset{i=0}{\prod }}G_{k+i}H_{k} & J_{s}\overset{P_{s}-1}{\underset{i=1}{\prod }}G_{s+i}H_{s+1} & \cdots & J_{s}\overset{P_{s}-1}{\underset{i=N_{c}-1}{\prod }}G_{s+i}H_{s+C_{s}-1} \end{array}\right] \end{array}$$

By solving Equation (22), the sequence of control inputs at each step is obtained, and the first element $\Delta \delta_{f}^{*}\left(s\right)\;$ in the sequence is chosen as the actual control variable:(24)$$\delta_{f}\left(s\right)=\delta_{f}\left(s-1\right)+\Delta \delta_{f}^{*}\left(s\right)$$

The above process is repeated at the next step.

## 5. Simulation Results and Analysis

To verify the proposed shared control framework, a series of cooperative emergency obstacle avoidance simulation studies were conducted jointly with CarSim-Matlab/Simulink. Parameters of the vehicle model used in CarSim are shown in Table 2. In this paper, the prediction horizon and control horizon of the proposed shared controller are $P_{s}=25$ and $C_{s}=$ 5, respectively, and the sampling time is $T_{s}=0.02\;s$. In addition, considering the physical limitations of the steering system, the constraints of the control input and the control increment are set as:$$\begin{array}{c} -10\;deg\leq \delta_{f}\leq 10\;deg\\ -0.85\;deg\leq \Delta \delta_{f}\leq 0.85\;deg \end{array}$$

To examine the robustness of the shared controller, three scenarios were designed with the varying of the driver’s steering commands, representing different driver’s states as described below:Distracted driving with no steering command;Obstacle avoidance in a panic with oscillating steering commands;Driver’s stochastic steering commands.

In these scenarios, the global reference trajectory is the centerline of the lane, and the local reference trajectory is the obstacle avoidance trajectory generated by the trajectory planning module. The proposed shared controller was tested at the speeds of 36 km/h, 72 km/h, and 108 km/h, respectively, in each scenario.

The sideslip angle, lateral acceleration and the heading angle deviation were used as indices to evaluate the performance of controllers. To verify the advantages of the smooth transition of the control authority, two controllers were designed based on the shared control framework proposed in the previous sections and simulated under the same conditions with only the strategy of authority allocation differing. Specifically, controller A conducted a smooth transition of control authority defined by Equation (16), while controller B conducted a step shift of control authority, which can be formulated as:(25)$$\sigma =\left\{\begin{array}{c} \begin{array}{cc} \;0 & ,TTC>TTC_{\max}\;and\;TLC>TLC_{\max} \end{array}\\ \begin{array}{cc} 1 & ,TTC\leq TTC_{\max}\;or\;TLC\leq TLC_{\max} \end{array} \end{array}\right.$$

### 5.1. Results and Analysis of Scenario 1

In this scenario, the vehicle is traveling on a road with a friction coefficient of 0.85, an obstacle appears at *X* = 150 m ahead, and the driver takes no action to avoid the obstacle.

As shown in Figure 5, Figure 6 and Figure 7, both controller A and controller B managed to steer the vehicle to avoid the obstacle successfully at different speeds, under the circumstance that the driver takes no steering action. Figure 5b, Figure 6b or Figure 7b clearly show that the shared controllers track the driver’s commands until the situation assessment module determines that the vehicle is in danger of collision and, hence, adjust the weights of the two objective terms in the MPC cost function. In the process of authority allocation in controller A, as marked as instance one in the figures, the front wheel angle changed more slowly and smoothly than that of controller B, which resulted in a smaller sideslip angle and lateral acceleration. Although controller A has full control later than controller B, the performance of controller A had no decrease compared with that of controller B, as there was no significant difference between the two controllers in the maximum sideslip angle and lateral acceleration during obstacle avoidance. Therefore, when the driver takes no action, a smooth transition of control authority can obtain better stability at different speeds.

### 5.2. Results and Analysis of Scenario 2

To verify the effectiveness of the proposed shared controller when the driver takes actions incorrectly, due to being flustered or immature driving skills, and turns the steering wheel left and right repeatedly. The driver’s steering command is simulated with a sine curve in this scenario.

As shown in Figure 8a, both controller A and controller B can help drivers keep the lane and avoid obstacles, under the circumstance that the driver takes oscillating steering actions. As shown in Figure 8c, the situation assessment module detected that the vehicle was about to deviate from the lane at instance 1, and adjusted the weights in the MPC cost function to keep the lane. When it was in danger of collision at instance 2, the driver steered the vehicle in the opposite direction of obstacle-avoidance. The controllers, therefore, set the objective to track the collision-free trajectory. Figure 8d,f show that the performance of controller B had a significant drop compared with that of controller A, as the peak values of the sideslip angle and the lateral acceleration of controller B were both higher than those of controller A. During the entire obstacle avoidance, the maximum sideslip angle and lateral acceleration of both controllers were no more than 1° and 0.2 g, respectively. Figure 9 and Figure 10 present the results of the same scenario, only with the vehicle speed increased to 72 km/h and 108 km/h, respectively. Again, it was shown that both controllers can assist the driver to complete the task of lane keeping and obstacle avoidance. However, when the risk of collision was detected at instance 2, the driver took the correct steering command, and controller A chose to track the driver’s command to avoid the obstacles while controller B ignored the driver’s steering commands and directly turned to track the reference trajectory. Therefore, it was proved that controller A attempted to track the driver’s command whenever it was safe and, consequently, improved the driver’s acceptance of automation without impacting the tracking performance.

### 5.3. Results and Analysis of Scenario 3

In scenario 3, the driver’s stochastic steering command was considered, and white noise was applied to simulate the driver’s steering command, which obeys standard normal distribution $N\left(0,1\right)$.

As shown in Figure 11, at different vehicle speeds, the driver’s steering command changed randomly and frequently. Both controller A and controller B could complete the task of lane keeping and obstacle avoidance. The robustness of the shared controllers to changes in driver’s inputs and vehicle speeds was proved. As shown in Figure 11b, the proposed shared controllers have the advantages of smoothing the driver’s steering commands. Moreover, also as shown in Figure 11b, the front wheel angle of controller B deviated from the driver’s command many times due to the risk of lane departure, while that of controller A deviated fewer times and was more consistent with the driver’s commands.

## 6. Conclusions

This paper presents a shared control strategy for lane keeping and obstacle avoidance in which the driver and the automation interact tightly at two levels, namely the tactical level and the operational level. At the tactical level, a trajectory planning controller based on NMPC generated the collision-free reference trajectories for the shared controller. A risk assessment algorithm was designed to determine the allocation of control authority, which considers the lateral and longitudinal risks simultaneously. At the operational level, a shared control framework based on MPC was proposed, whose cost function consists of two objective terms representing the objective to track reference trajectories and match the driver’s steering commands, respectively. A smooth combination of control actions between the driver and the automation can be realized in an optimal controller by adjusting the weights of the objective terms, rather than changing the weights of the control commands of the driver and the automation. The simulation results show that the proposed control framework can robustly assist the driver to complete the task of lane keeping and obstacle avoidance when the vehicle velocity and the driver’s steering command varies, while maintaining vehicle stability. It was also discovered that, as compared to a step shift of control authority, a smooth transition of control authority can further improve the performance while offering more freedom to the driver. Future research should attempt to explore more human factors in the design of the shared control scheme to improve the driver’s acceptance of automation.

## Figures and Tables

**Figure 1 sensors-21-04671-f001:**
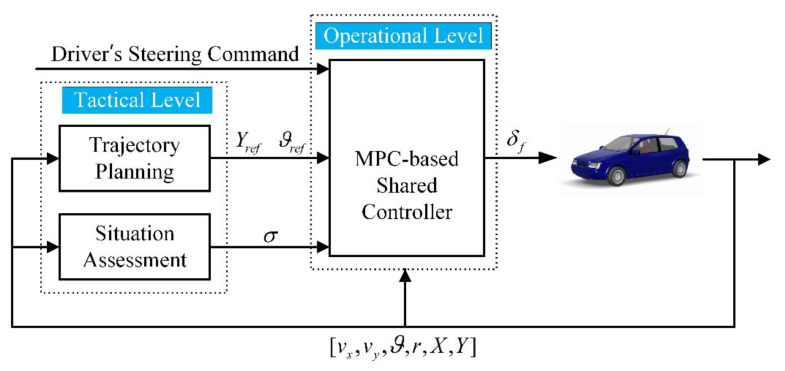
The framework of the proposed shared control scheme.

**Figure 2 sensors-21-04671-f002:**
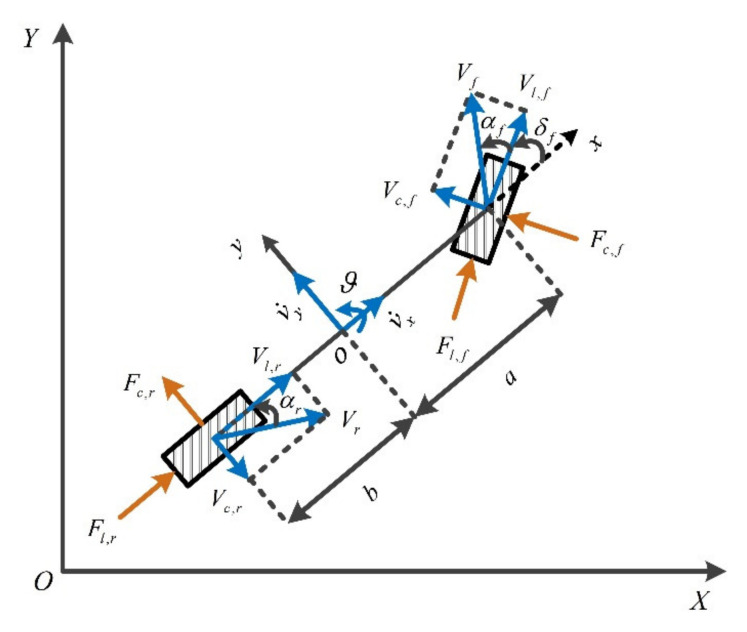
Vehicle dynamics model.

**Figure 3 sensors-21-04671-f003:**
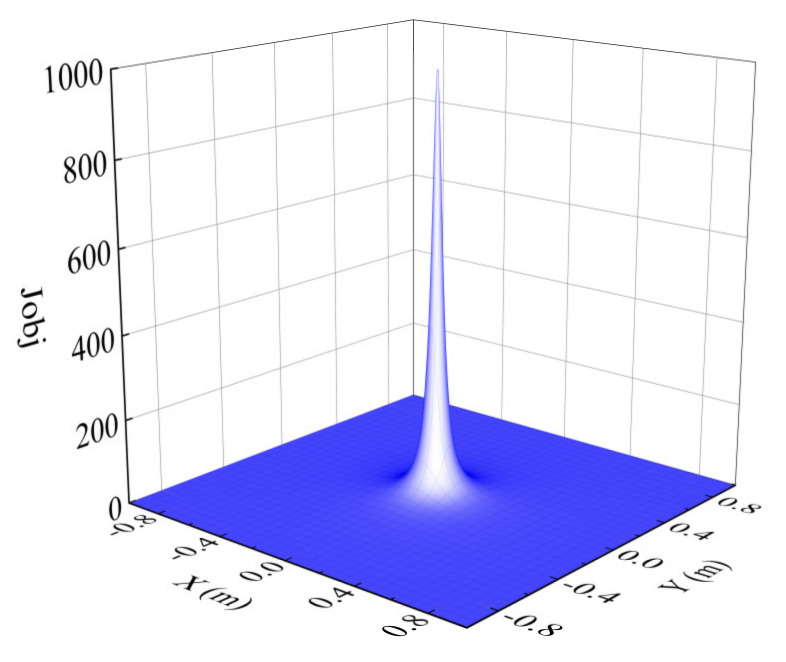
Obstacle avoidance function.

**Figure 4 sensors-21-04671-f004:**
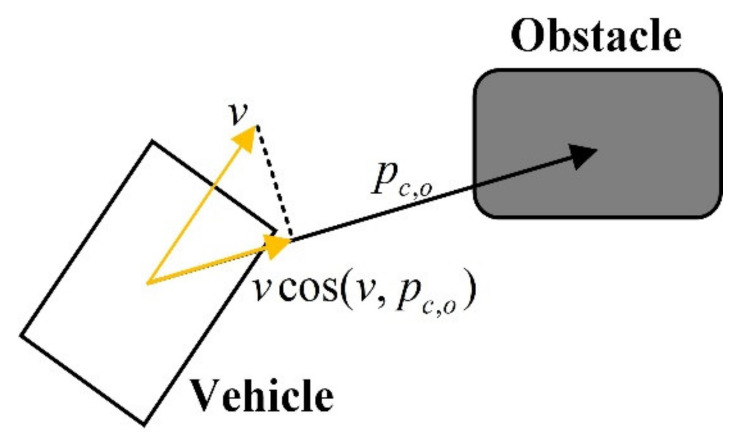
Diagram of the calculation of TTC.

**Figure 5 sensors-21-04671-f005:**
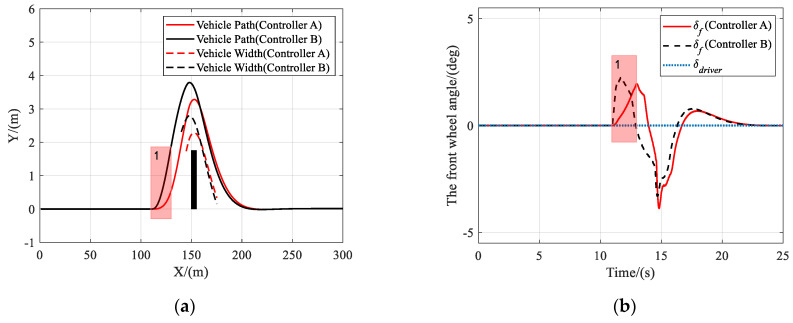
Scenario 1: Simulation results at 36 km/h. (**a**) Vehicle trajectory, (**b**) the actual front wheel angle, (**c**) sideslip angle, and (**d**) lateral acceleration.

**Figure 6 sensors-21-04671-f006:**
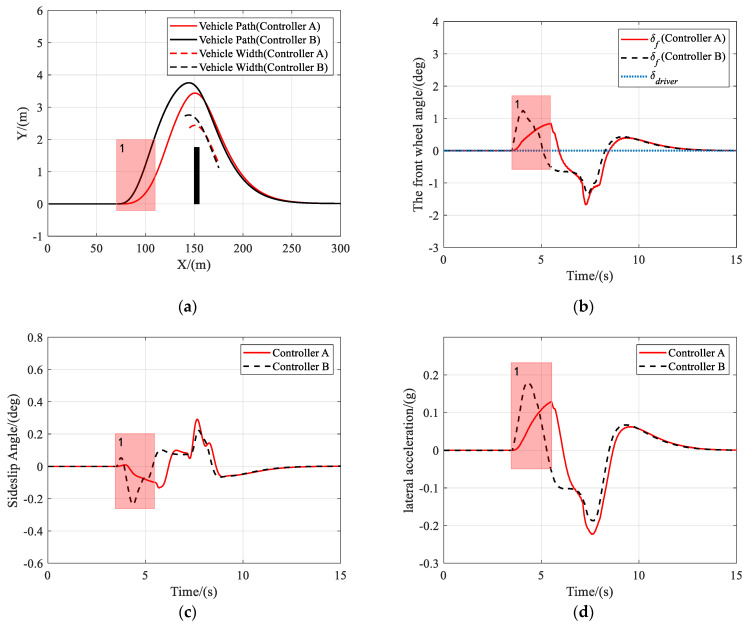
Scenario 1: Simulation results at 72 km/h. (**a**) Vehicle trajectory, (**b**) the actual front wheel angle, (**c**) sideslip angle, and (**d**) lateral acceleration.

**Figure 7 sensors-21-04671-f007:**
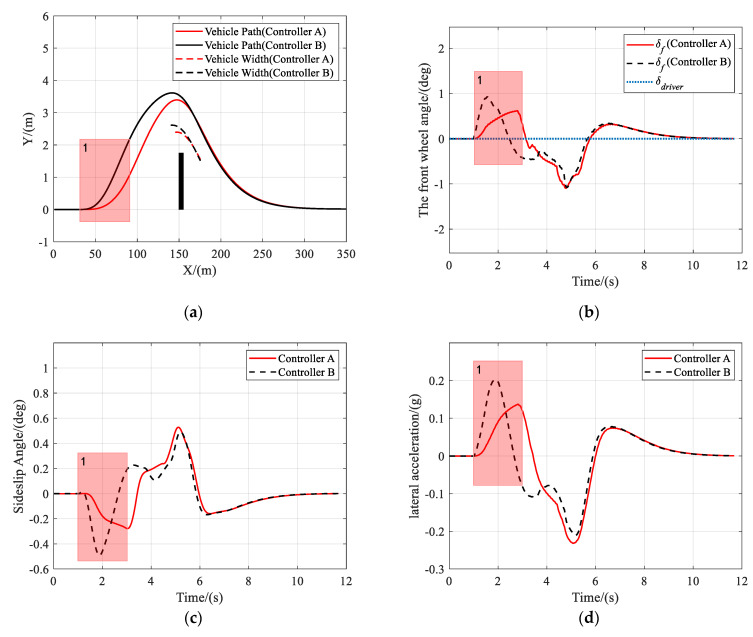
Scenario 1: Simulation results at 108 km/h. (**a**) Vehicle trajectory, (**b**) the actual front wheel angle, (**c**) sideslip angle, and (**d**) lateral acceleration.

**Figure 8 sensors-21-04671-f008:**
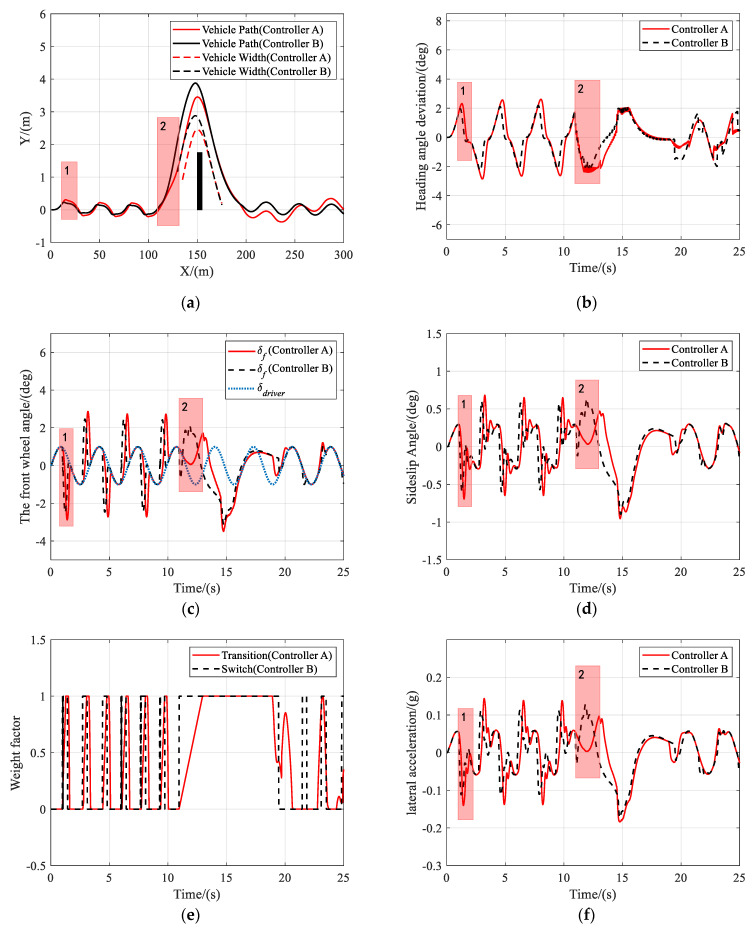
Scenario 2: Simulation results at 36 km/h. (**a**) Vehicle trajectory, (**b**) heading angle deviation, (**c**) the front wheel angle, (**d**) sideslip angle, (**e**) weight factor, and (**f**) lateral acceleration.

**Figure 9 sensors-21-04671-f009:**
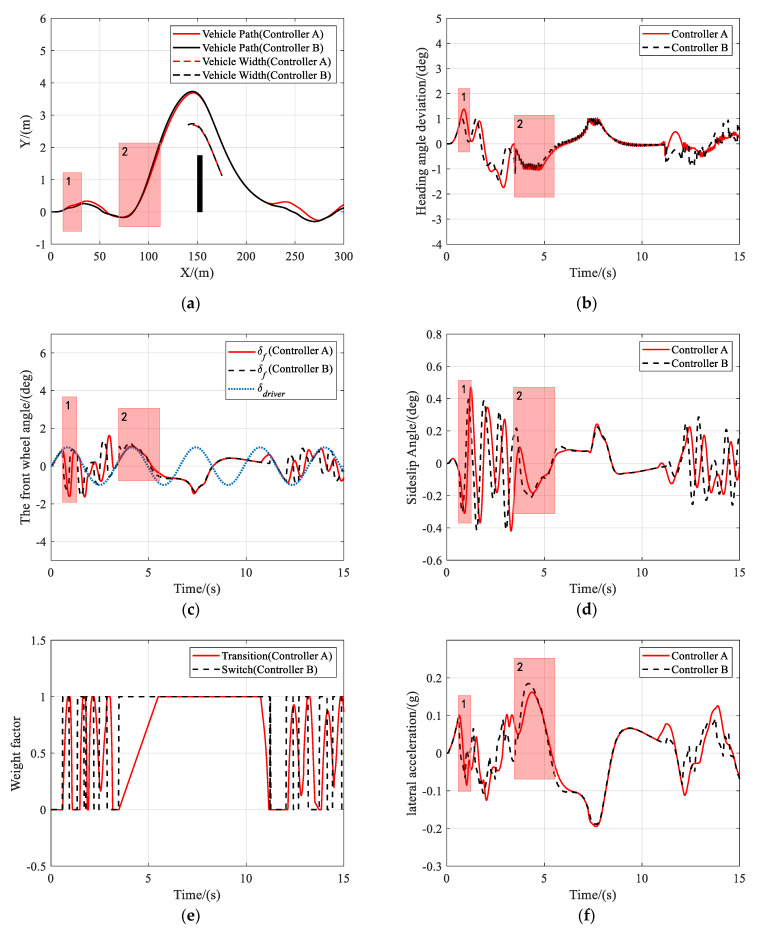
Scenario 2: Simulation results at 72 km/h. (**a**) Vehicle trajectory, (**b**) heading angle deviation, (**c**) the front wheel angle, (**d**) sideslip angle, (**e**) weight factor, and (**f**) lateral acceleration.

**Figure 10 sensors-21-04671-f010:**
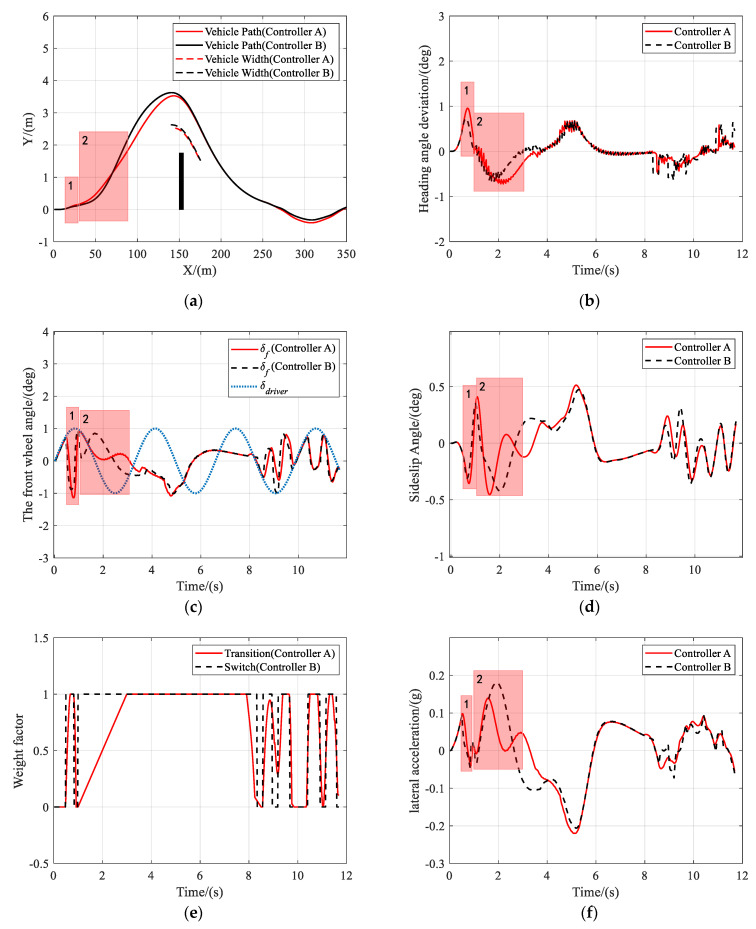
Scenario 2: Simulation results at 108 km/h. (**a**) Vehicle trajectory, (**b**) heading angle deviation, (**c**) the front wheel angle, (**d**) sideslip angle, (**e**) weight factor, and (**f**) lateral acceleration.

**Figure 11 sensors-21-04671-f011:**
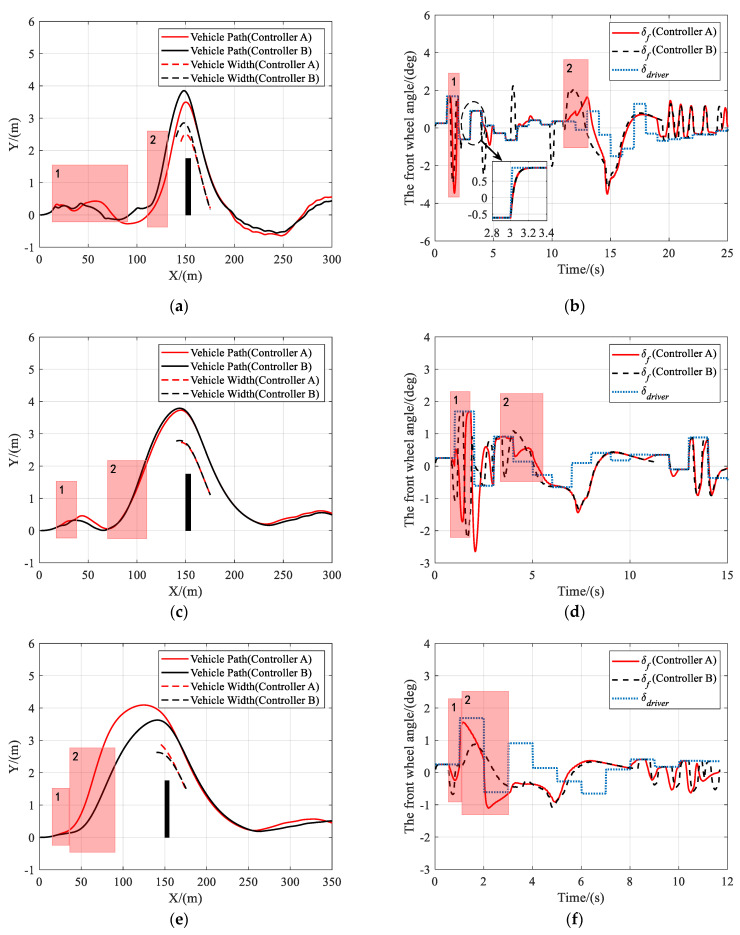
Scenario 3: Simulation results at different speed. (**a**) Vehicle trajectory at 36 km/h, (**b**) the front wheel angle at 36 km/h, (**c**) vehicle trajectory at 72 km/h, (**d**) the front wheel angle at 72 km/h, (**e**) vehicle trajectory at 108 km/h, and (**f**) the front wheel angle at 108 km/h.

**Table 1 sensors-21-04671-t001:** Vehicle model parameters.

Symbol	Description
m	Vehicle mass
$I_{z}$	Vehicle inertia
a/b	Distance from front/rear axle to center of mass
$C_{c,f}/C_{c,r}$	Cornering stiffness of front/rear tires
$C_{l,f}/C_{l,r}$	Lateral stiffness of front/rear tires
$\alpha_{f}/\alpha_{r}$	Slip angle of front/rear tires
$\kappa_{f}/\kappa_{r}$	Longitudinal slip ratio of front/rear tires
$F_{c,f}/F_{c,r}$	Lateral force of front/rear tires
$F_{l,f}/F_{l,r}$	Longitudinal force of front/rear tires
$\delta_{f}$	Front wheel steering angle
$v_{x}$	Vehicle longitudinal velocity
$v_{y}$	Vehicle lateral velocity
$\gamma_{y}$	Lateral acceleration
ϑ	Vehicle heading angle
r	Yaw rate
X	Vehicle longitudinal position in inertial coordinate
Y	Lateral position of vehicle in inertial coordinate

**Table 2 sensors-21-04671-t002:** Parameters of the vehicle model.

Symbol	Value
m	1723 kg
$I_{z}$	4175 kg·m^2^
a	1.232 m
b	1.468 m

## Data Availability

Not applicable.

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
