# Peer review of "Shared Steering Control for Lane Keeping and Obstacle Avoidance Based on Multi-Objective MPC"

_sensors, 2021, doi:10.3390/s21144671_

Round 1

Reviewer 1 Report

This paper studies the steering control authority problem between a driver and an automation system. The authors solved the problem using multi-objective MPC. If the authors proposed the method for the future driver assistance system or active safety system in the autonomous driving research field. The paper explains the purpose and the scope of the research clearly. However, the application of the MPC for the steering control of autonomous vehicle has been studied by numerous researchers and the reviewer found that the originality of the paper is not significant. The authors evaluated the performance of the algorithm using CarSim simulation. However, the implementation of the control algorithm is not clearly presented and the advantages obtained by using MPC is not explained in detail.

The detailed review comments are as follows,

  1. The major advantages of using Model Predictive Control is its ability to employ the constraints during the optimization process. But this paper does not describe the benefits of using MPC in detail. In my opinion, the paper needs to mention MPC controller operating environments like receiving horizon, control period, constraints and other parameters in the simulation results.
  2. One of the main design points of MPC is the maximum computational time that occurs during optimization. The authors need to analyze the computation time.
  3. In figure 4. Chinese characters should be changed to English.
  4. When the paper is printed grayscale, the line is difficult to distinguish at the graphs that the controller A, controller B, driver command is included. If you change one of the solid lines to another, the visibility will improve.
  5. Simulation test analysis does not seem sufficient to show the superiority of this algorithm. In particular, analyzing only the starting point of obstacle avoidance is not sufficient evidence. The detailed explanation is necessary about how the Control A and Control B are designed.
  6. The references are not properly organized. For example, the 12th reference does not contain information about the author.

Reviewer 2 Report

The paper deals with the shared steering control framework for lane keeping and obstacle avoidance based on multi-objective model predictive control. Although the paper is easy to follow, the content is questionable - reasons for which are given below.

  • MPC based approaches for shared lane keeping /obstacle avoidance control are well known in the literature. The online adaptation of the weight in the cost function in order to give the authority to the driver or to the system is already proposed in the work of Guo et al. (2017). Please, at least include remarks about the main difference with these approaches.

[Guo C., Sentouh C., Popieul J.-C., Haué J.-B. (2017). MPC-based Shared Steering Control for Automated Driving Systems. IEEE International Conference on Systems, Man, and Cybernetics, October.

  • In line 200 : please correct the sentence as : Where, ???min and ?T?min represent the low-risk threshold of ??? and ???, ?L?max and ???max represent the high-risk threshold of ??? and ???, respectively.

I think that the TLC is not a good index to activate and deactivate the shared control because it is difficult to guaranties the acceptability of the designed system by the driver… the TLC based risk threshold is very subjective…  

  • In addition, the TLC formula in equation (13) depends on the lateral vehicle speed and it is well known that this variable is not available for measurement using a conventional sensor. How you plan to implement the proposed TLC algorithm in real application with conventional sensors.
  • When we deal with shared control problems in automated driving, one of the most challenging problems to resolve is the management of the conflict between the driver and the lane-keeping system. In your system, what happens if the driver tries to change lanes in the case where there is an object on the lane that has not been detected by the system vision?
  • The acceptability of the controller is crucial in a shared context: however, the paper seems to test the system with a numerical simulation. This is quite limiting and must be further developed.
  • The authors should explain, in the introduction, what is the main contribution of this paper compared to the existing multi-objective MPC based shared control for lane keeping and obstacle avoidance? The authors should point their work in relation to others. Make clear the survey and report the advances of the proposed approach. The connection and contribution of this paper should be stated in a more transparent way in the introduction part. Please explain more in detail the problem that the paper tries to resolve.

Round 2

Reviewer 1 Report

The authors revised the manuscript according to the suggestions provided by the previous review. The details of how MPC is implemented and the advantages of the MPC are well explained in the revised paper.

Reviewer 2 Report

All of the concerns in the first review have been nicely answered by the authors. I recommend  the paper for publication.